# Articular cartilage corefucosylation regulates tissue resilience in osteoarthritis

**Kentaro Homan[1], Tomohiro Onodera[1]\*, Hisatoshi Hanamatsu[2], Jun-ichi Furukawa[1,2], Daisuke Momma[3], Masatake Matsuoka[1], Norimasa Iwasaki[1]**

[1]Department of Orthopaedic Surgery, Faculty of Medicine and Graduate School of Medicine, Hokkaido University, Sapporo, Japan; [2]Institute for Glyco-core Research (iGCORE), Nagoya University, Nagoya, Japan; [3]Center for Sports Medicine, Hokkaido University Hospital, Sapporo, Japan

**Abstract** This study aimed to investigate the glycan structural changes that occur before histological degeneration in osteoarthritis (OA) and to determine the mechanism by which these glycan conformational changes affect cartilage degeneration. An OA model was established in rabbits using mannosidase injection, which reduced high-mannose type N-glycans and led to cartilage degeneration. Further analysis of glycome in human OA cartilage identified specific corefucosylated N-glycan expression patterns. Inhibition of N-glycan corefucosylation in mice resulted in unrecoverable cartilage degeneration, while cartilage-specific blocking of corefucosylation led to accelerated development of aging-associated and instability-induced OA models. We conclude that α1,6 fucosyltransferase is required postnatally to prevent preosteoarthritic deterioration of articular cartilage. These findings provide a novel definition of early OA and identify glyco-phenotypes of OA cartilage, which may distinguish individuals at higher risk of progression.

## eLife assessment

This **valuable** study sheds light on the pivotal role of alterations in chondrocyte glycan metabolism in two contexts: The onset of cartilage degeneration and early onset of osteoarthritis (OA). The action is through hypertrophic differentiation of chondrocytes, a finding that provides insights into the identification of nascent markers for early-stage OA. The evidence supporting the claims is **solid**, with the authors clearly demonstrating the role of articular cartilage corefucosylation in the development of OA. The authors' inferences would be further enhanced through future experiments aimed at analyzing the mechanisms underlying the changes in glycometabolism in cartilage.

## Introduction

Osteoarthritis (OA), a degeneration of joint components, such as proteoglycans (PGs) and collagen, destroys cartilage and impairs joint function (*Grillet et al., 2023*). The global prevalence of OA is estimated to be >500 million, which is a significant socioeconomic burden (*Safiri et al., 2020*). Therefore, preventive measures must be implemented to reduce the future disease burden and disability. Once the cartilage is destroyed, it cannot be repaired homogeneously and progresses irreversibly thereafter (*Thorstensson et al., 2009*; *Huey et al., 2012*). Since articular cartilage has no blood vessels and is not innervated, it is difficult for symptoms to appear. By the time clinical symptoms appear, the condition is often already considerably advanced (*Glyn-Jones et al., 2015*). Therefore, from a preventive perspective, it is necessary to define and detect 'pre-OA', the stage before the onset of early OA (*Chu,*

**\*For correspondence:**
tomozou@med.hokudai.ac.jp

**Competing interest:** The authors declare that no competing interests exist.

2016; *Mahmoudian et al., 2021*). Pre-OA is a knee with no pain, no motor function problems, and no arthroscopic or structural changes on radiography, only emerging cellular effects. Despite many studies, there is no gold standard for OA biomarkers (*Henrotin, 2022*). It is important to characterize the disease state before OA to shift the paradigm from mitigation to late-onset disease prevention.

In the very early stages of OA, before cartilage matrix degeneration, cartilage destruction by enzymatic processes that degrade glycoproteins, PGs, and collagen, in that order, is underway (*Vignon et al., 1987*). Collagenolytic activity is largely unchanged on intact articular surfaces (*Brew et al., 2010*; *Otero, 2012*), and glycoprotein degradation is thought to be an early transient event in the enzymatic process of cartilage destruction because hexose degradation of glycoproteins promotes degradation of the protein portion of the molecule by proteinases (*Goldring, 2012*). Additionally, cartilage damage is reversible if type 2 collagen breakdown is mild (*van Meurs et al., 1999*). Structural change analysis in glycoproteins is necessary for the early stages of OA but has not yet been performed sufficiently.

Matsuhashi et al.'s study showed that structural changes in N-glycans occurred in degenerated cartilage in a rabbit OA model before tissue changes occurred (*Matsuhashi et al., 2008*). *Urita et al., 2011* demonstrated that these N-glycans in degenerated mice and human cartilage are high-mannose type N-glycans. Furthermore, among several glycosidases, mannosidase activity was very high in OA joints, and its activity was high before the progression of cartilage fibrosis (*Snelling et al., 2014*). This suggests that high-mannose type N-glycan plays an important role in OA pathogenesis. However, whether changes in the glycan structure are involved in OA pathogenesis is unclear.

This study analyzed the glycan structural changes that preceded OA degeneration and elucidated the mechanism of glycan structural changes in cartilage degeneration. For this purpose, the effects of mannosidase, a specific degrading enzyme of high-mannose type N-glycans associated with OA, on cartilage tissues and its molecular mechanisms were analyzed.

## Results

### Mannosidase injection caused early OA-like change in vivo

A decrease in Safranin O staining in the right femoral condyle of the α-mannosidase-injected rabbit knee after 4 weeks was observed (*Figure 1A and B*). Terminal deoxynucleotidyl transferase dUTP nick end labeling (TUNEL) staining was enhanced in the superficial layer compared with that in control. Articular cartilage chondrocytes exposed to α-mannosidase were not arranged in columns in the middle zone. The surfaces of each group were smooth and had a good macroscopic appearance (*Figure 1C*). None of the groups showed a decrease in type 2 collagen staining, and the orientation was maintained (*Figure 1D*). To achieve optimal activity, the pH range of 4–5 is ideal for mannosidase. In a physiological saline solution with a pH of 7 and a concentration of 100 µg/ml, it exhibited specific activity. Nevertheless, at a concentration of 1.0 µg/ml, this activity was no longer present (*Figure 1—figure supplement 1*). The mean histological score of the α-mannosidase group was 0.83, indicating mild OA (*Figure 1E*). The control group was unchanged. Cartilage degeneration during intermittent intra-articular administration of mannosidase was reversed upon discontinuation (*Figure 1F*).

### Depletion of high-mannose type N-glycans in chondrocytes leads to cartilage degradation ex vivo

To quantify the degeneration and resilience promoted by the injection of α-mannosidase, the cultured cartilage in mice was harvested at different time points (*Figure 2A*). Concanavalin A (Con A) reactivity declined in articular cartilage treated with α-mannosidase (*Figure 2B*). The application of α-mannosidase stimulated the release of PG from cultured cartilage in association with extracellular matrix degradation (*Figure 2C*) and showed significantly higher spontaneous nitric oxide (NO) release than the control over time (*Figure 2F*). Mannosidase-exposed cartilage showed an increase in TUNEL-positive cells in chondrocytes in the deep zone and type 10 collagen-positive cells in superficial chondrocytes; however, these changes did not occur in cytokine-stimulated cartilage (*Figure 2—figure supplement 1*). Mannosidase exposure increased *Adamts5* expression and suppressed *Mmp13* expression and anabolic factors (*Figure 3E*). In the articular cartilage that was exposed to freeze-thaw treatment before culture, no differences in PG loss were observed between the α-mannosidase-stimulated cartilage and non-stimulated cartilage (*Figure 2E*). Excluding mannosidase loading, PG leakage and

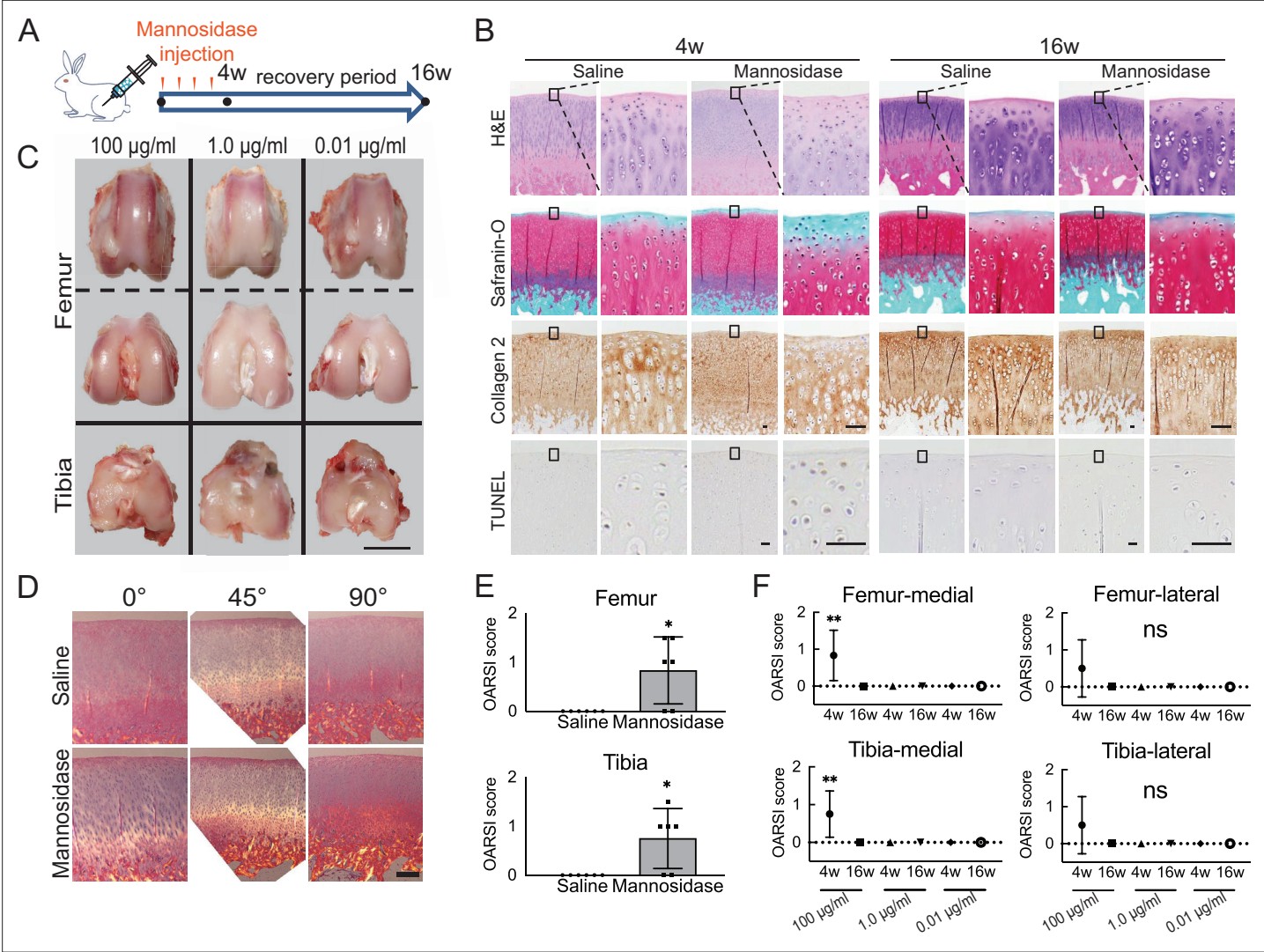

**Figure 1.** Effects of α-mannosidase injection in the knee joint in vivo. (**A**) Time course of intra-articular α-mannosidase injection and unloading. Red arrows indicate the one-shot, intra-articular injection that was performed once a week. (**B**) Histological evaluation of knee articular cartilage 4 weeks after intermittent α-mannosidase infusion and 16 weeks after load removal. Scale bars, 50 μm. (**C**) Representative macroscopic assessment of the articular cartilage using mannosidase concentration. Scale bar, 1 cm. (**D**) Evaluation of the collagen orientation of reparative tissues at 4 weeks postoperatively. Sections stained with HE was viewed under a polarized light microscope at multiple angles (0°, 45°, and 90°). Scale bar, 200 μm. (**E**) OARSI scores at 4 weeks postoperatively on sections from the knee joint (n=6). (**F**) Tissue degeneration scores at each time point are based on mannosidase concentration. Data are shown as mean ± standard deviation. *p<0.05, **p<0.01 versus the saline group in (**E**), and versus 4 weeks in (**F**). In (**E**), the Welch t-test was used for statistical analysis. In (**F**), n=6 rabbits per group. One-way analysis of variance with Tukey's multiple comparison test was used to perform statistical analysis. HE, hematoxylin and eosin; OARSI, Osteoarthritis Research Society International; TUNEL, terminal deoxynucleotidyl transferase dUTP nick end labeling.

The online version of this article includes the following figure supplement(s) for figure 1:

**Figure supplement 1.** Enzymatic activity under physiologic pH conditions in joint injection and cartilage culture systems.

---

Safranin O staining were restored (*Figure 2D*), but Con A staining loss was not. Sialidase addition did not cause PG leakage or NO production (*Figure 2—figure supplement 2*), and the aggrecan lost in the catabolic phase was not replenished in the subsequent anabolic phase.

## Corefucosylated N-glycan was formed in resilient cartilage and isolated chondrocyte

Glycoblotting analysis revealed a significant difference in the N-glycan profiles of mouse cartilage before and after mannosidase treatment (*Figure 3A* and *Table 1*). Plotting m/z after mannosidase

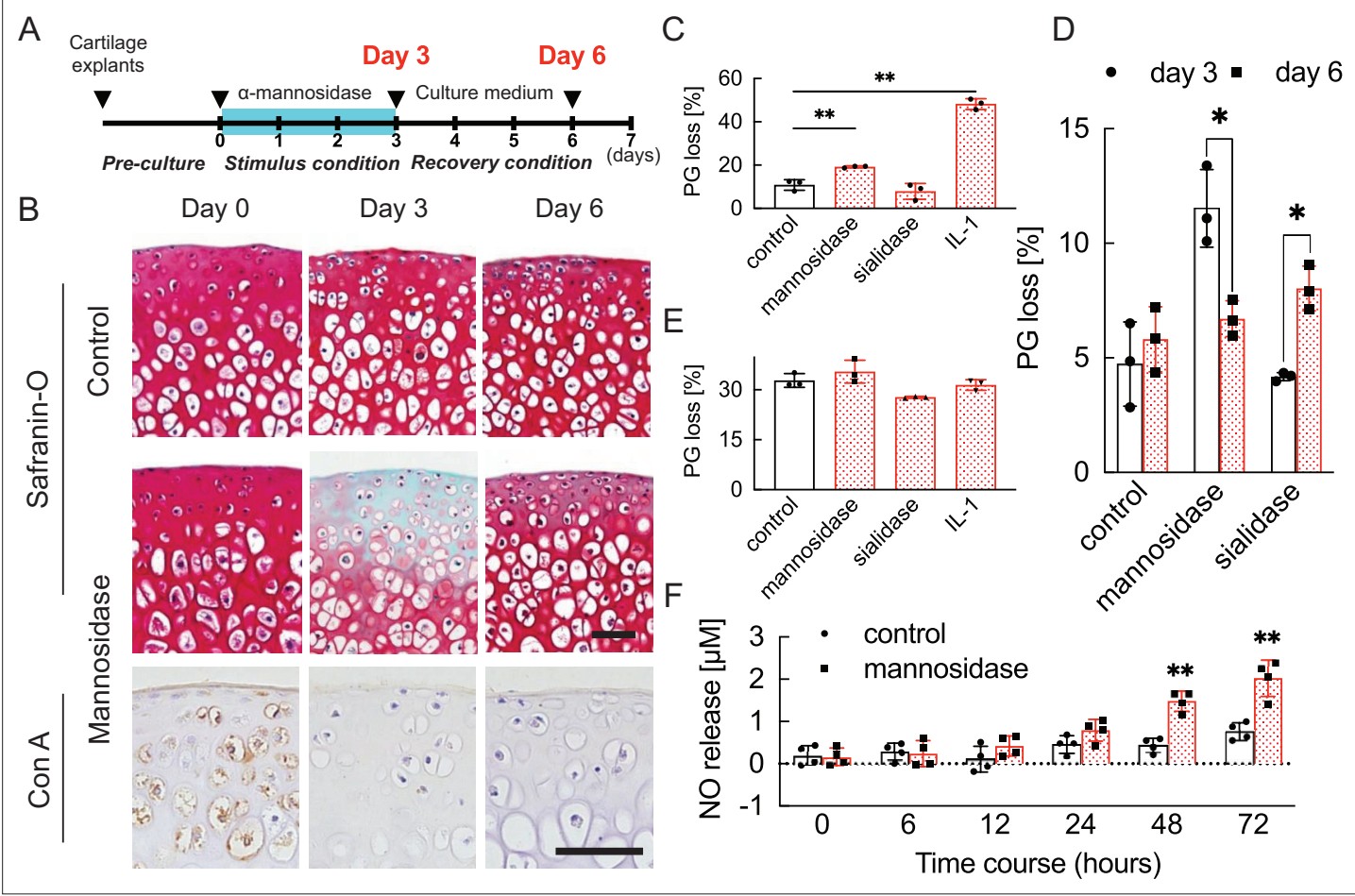

**Figure 2.** Cartilage degradation induced by mannosidase stimulation. (**A**) Schematic diagram of the procedures used to establish the early OA model before irreversible and progressive destruction of articular cartilage. (**B**) Mannosidase-treated cartilage stained with Safranin O and Con A. The distribution of high-mannose type N-glycans is decreased in mannosidase-induced degraded mouse cartilage. Scale bars, 50 μm. (**C**), (**D**) PG release after exposure to mannosidase (**C**) and recovery after its removal from cultured cartilage grafts (**D**). (**E**) To investigate the non-chondrocyte-mediated release of fragments, PG loss was measured in cartilage explants that had undergone freeze-thaw cycles. (**F**) Sequential NO released into media measured as concentrations of nitrite for cartilage explants stimulated with mannosidase. Data are shown as mean ± standard deviation (s.d.). *p<0.05, **p<0.01 versus the control group in (**C**), (**E**), and (**F**), and versus day 3 in (**D**). In (**C**) and (**E**), n=3 samples (six mice) per group. One-way ANOVA with the Tukey multiple comparison test was used to perform statistical analysis. In (**D**), n=12 mice in each group and n=3 samples (six mice) at each time point. In (**F**), n=16 mice at each time point and n=4 samples (eight mice) per group. In (**D**) and (**F**), two-way ANOVA with the Sidak multiple-comparisons test was used to perform statistical analysis. Con A, concanavalin A; NO, nitric oxide; ANOVA, analysis of variance; PG, proteoglycan.

The online version of this article includes the following figure supplement(s) for figure 2:

**Figure supplement 1.** Histological effects of mannosidase on cartilage culture systems.

**Figure supplement 2.** Impact of other glycoside hydrolases on NO production from cartilage.

exposure against m/z before mannosidase exposure demonstrated that only N-glycans were affected by enzyme dissociation from the y=x line (*Figure 3B*, *Table 2*). In both denatured and reverted cartilages, Man8-9GlcNAc2 was decreased, and Man5-6GlcNAc2 was increased by the action of mannosidase, which has a high specificity for the terminal mannosidic residues of glycans (*Figure 3C and D*). In contrast, corefucosylated N-glycans, which were formed in the same biosynthesis pathway, were also found (*Figure 3D*). Similarly, the N-glycan profiles of isolated chondrocytes showed the presence of this corefucosylated structure (*Figure 3—figure supplement 1*). The expression of *Fut8*, the only enzyme involved in corefucosylation, increased significantly in mannosidase-treated cartilage (*Figure 3E*).

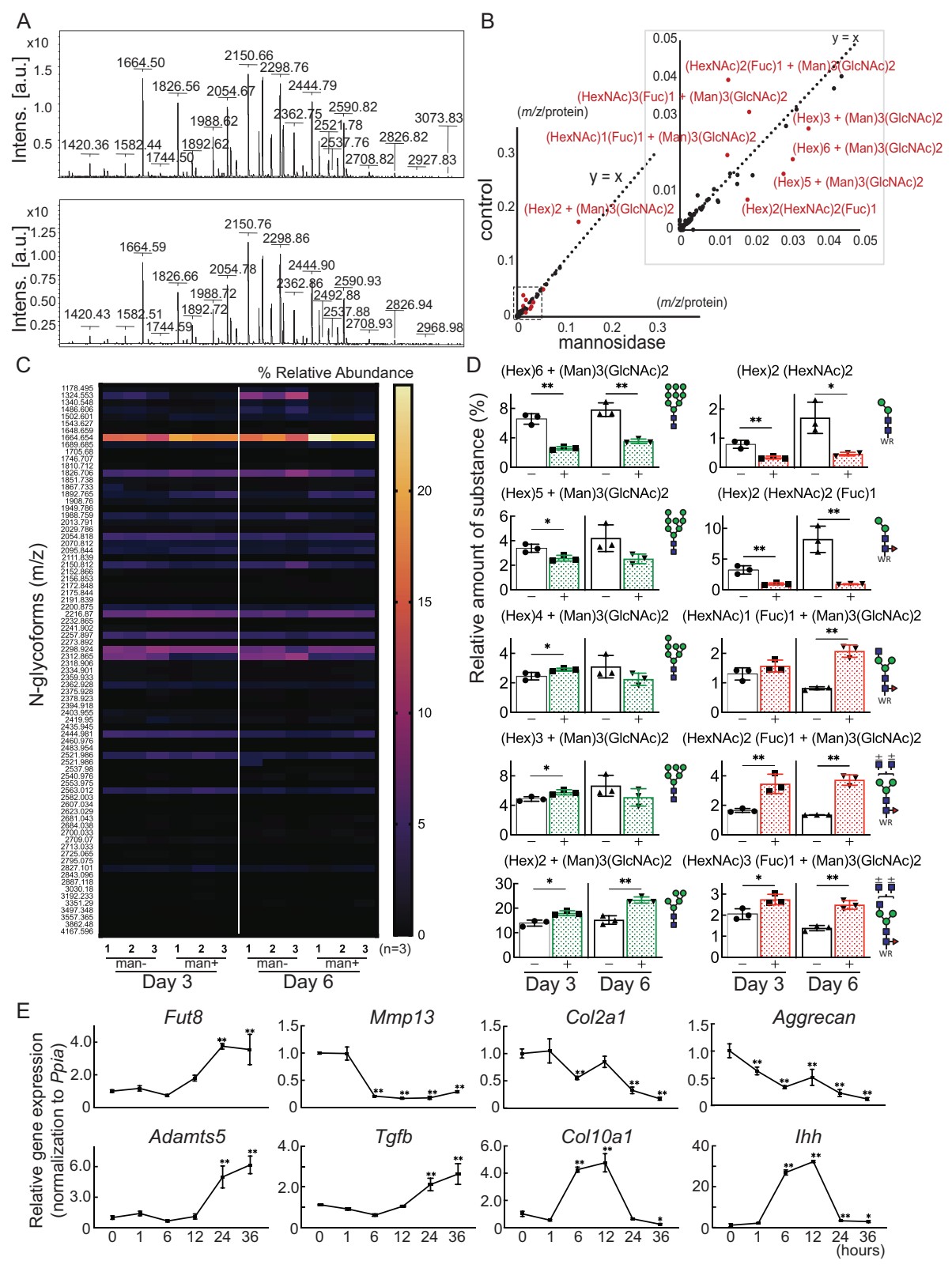

**Figure 3.** Cartilage N-glycomes. (**A**) MALDI-TOF MS spectra showing the quantitative N-glycan profiles of cartilage before (upper) and after (lower) mannosidase treatment. (**B**) Scatter plot of the changes in *m/z* per protein abundance in the mannosidase (x-axis) and control (y-axis) N-glycan structures. The red dots indicate the top 10 N-glycan structures based on the deviation from the y=x line. (**C**) Heatmap of N-glycome showing differential N-glycan expression in the mannosidase (man+) group versus the control (man-) group. (**D**) N-glycans altered by mannosidase were

*Figure 3 continued on next page*

*Figure 3 continued*

corefucosylated glycans and high-mannose type N-glycans. (**E**) RT-PCR expression analysis of marker genes in organ culture of cartilage with mannosidase. Data are shown as mean ± standard deviation. *p<0.05, **p<0.01 versus the control group in (**D**), and versus 0 hours in (**E**). In (**D**) and (**E**), n=3 samples (six mice) per group. In (**D**), unpaired t-tests were used to perform statistical analyses. In (**E**), one-way ANOVA with the Dunnett multiple comparison test was used to perform statistical analysis. MALDI-TOF MS, matrix-assisted laser desorption/ionization-time of flight mass spectrometry; ANOVA, analysis of variance; RT-PCR, real-time polymerase chain reaction.

The online version of this article includes the following figure supplement(s) for figure 3:

**Figure supplement 1.** Glycoform present in the N-glycans of chondrocytes: core fucose.

## Deficiency of corefucosylated glycans in articular cartilage inhibits recovery from cartilage damage and promotes cartilage degeneration

We first examined whether the absence of core fucose affects the resilience of cartilage degeneration. Mice lacking *Fut8* exhibited severe growth retardation and died shortly after birth. Therefore, we decided to generate *Fut8* conditional knockout (cKO) mice (*Figure 4A and B*). The tissue specificity of recombination was confirmed by polymerase chain reaction (PCR) analysis of mouse genomic DNA that detected either the intact *Fut8*^loxP allele or the Cre-recombined *Fut8*^Δ allele in repeated crosses; N-glycan analysis of cartilage demonstrated the loss of corefucosylated sugar chains (*Figure 4C*). The proportion of high-mannose type N-glycan content was unchanged in cartilage lacking the *Fut8* gene (*Figure 4—figure supplement 1*). Next, we used real-time (RT)-PCR to determine relative *Fut8* messenger (m)-RNA levels in the cartilage of *Col2a1;Fut8*^−/− mice and their wild-type (WT) littermates (*Fut8*^loxP/loxP). Real-time polymerase chain reaction (RT-PCR) analysis showed that *Fut8* mRNA expression was reduced by >99% in the articular cartilage of *Col2a1;Fut8*^−/− mice compared to that in their WT littermates (p<0.01; n=3 mice per group) (*Figure 4D*). Chondrocyte-surface corefucosylation levels were analyzed by lectin staining using *Pholiota squarrosa* lectin (PhoSL), which specifically recognizes α1–6 core fucose (*Kobayashi et al., 2012*). PhoSL lectin staining was completely absent in the articular cartilage of *Fut8* cKO mice, in contrast to the positive staining observed in *Fut8* flox mice under physiological conditions (*Figure 4E*). Enzymatic degradation of high-mannose type N-glycans on the cartilage in *Fut8*-deficient mice exacerbated cartilage degeneration (*Figure 4E and F*). In addition, reversible matrix repair by enzyme removal observed in control cartilage was not observed in *Fut8*-deficient cartilage explant cultures. To evaluate whether TGF-β1 decreases cartilage degeneration after mannosidase stimulation, TGF-β1 was exogenously added to *Col2a1;Fut8*^−/− cartilage in the presence of α-mannosidase stimulation for 24 hr. The samples treated with TGF-β1 leaked significantly less PG following mannosidase stimulation compared to samples not treated with TGF-β1 (*Figure 4F*).

## Loss of corefucosylated glycans in articular cartilage leads to premature OA

*Col2a1;Fut8*^−/− mice developed and grew normally without major organ abnormalities of major organs and could not be distinguished from their WT littermates. The whole skeletons of newborn mice stained with alcian blue and alizarin red did not differ in appearance between the genotypes (*Figure 5A*). *Col2a1;Fut8*^−/− mice showed a lower body weight than their WT littermates, as indicated by the growth curve (*Figure 5B*). To assess the role of FUT8 in endochondral ossification, we performed an epiphyseal plate analysis of 4-week-old *Col2a1;Fut8*^−/− mice. This uncovered a significant enlargement of the zone of hypertrophic chondrocytes in the growth plates of the long bones of *Col2a1;Fut8*^−/− mice compared to controls (*Figure 5C*, *Figure 5—figure supplement 1*). We analyzed the development of instability-induced OA changes in *Fut8*^loxP/loxP and *Fut8* cKO mice. Sham operations produced no significant changes in OA in either mouse genotype. Eight weeks after instability-inducing surgery, the joints of WT mice developed changes in OA, such as cartilage erosion and a reduction in Safranin O staining and chondrocyte number. Deletion of FUT8 results in more severe changes in OA. In the joints of *Fut8* cKO mice, 8 weeks after surgery, the noncalcified zone (the upper layer above the tidemark) was almost completely lost. The number of chondrocytes was significantly reduced (*Figure 5D*). Quantitative assessment using the OA Research Society International (OARSI) score supported these histological findings (difference: –2.900; *Fut8*^flox/flox/destabilization of the medial meniscus (DMM) versus *Col2a1-Cre; Fut8*^flox/flox/DMM, 95% confidence interval [CI] for the difference: –5.027 to –0.7734) (*Figure 5E*). Changes in OA progressed earlier in the aging models of *Fut8*-deficient

**Table 1.** N-glycans affected by α-mannosidase.

Nineteen of 78 glycans underwent significant changes on either day 3 or 6.

| | Average | | | | SD | | | | p-value | |
|---|---|---|---|---|---|---|---|---|---|---|
| | day 3 | | day 6 | | day 3 | | day 6 | | | |
| | man- | man+ | man- | man+ | man- | man+ | man- | man+ | day 3 | day 6 |
| (Hex)2 (HexNAc)2 | 0.792357 | 0.338717 | 1.698703 | 0.456702 | 0.13679 | 0.050814 | 0.535795 | 0.06191 | 0.005751 | 0.016286 |
| (Hex)2 (HexNAc)2 (Fuc)1 | 3.234707 | 0.961793 | 8.220451 | 0.924848 | 0.674962 | 0.171418 | 2.152247 | 0.065893 | 0.004824 | 0.00421 |
| (Hex)3 (HexNAc)2 (Fuc)1 | 1.490549 | 0.803672 | 2.709981 | 1.15013 | 0.296407 | 0.119276 | 0.737097 | 0.183631 | 0.020411 | 0.023657 |
| (Hex)4 (HexNAc)2 | 0.835177 | 1.59262 | 1.387137 | 1.927679 | 0.149341 | 0.226128 | 0.39749 | 0.261157 | 0.008393 | 0.120372 |
| (Hex)4 (HexNAc)2 (Fuc)1 | 0.376984 | 0.283877 | 0.54256 | 0.352081 | 0.044946 | 0.018371 | 0.106303 | 0.060982 | 0.029345 | 0.054541 |
| (Hex)2 + (Man)3(GlcNAc)2 | 13.90417 | 17.95947 | 15.15816 | 23.33117 | 1.234223 | 1.006214 | 1.724106 | 1.244995 | 0.011592 | 0.002645 |
| (HexNAc)1 (Fuc)1 + (Man)3(GlcNAc)2 | 1.301597 | 1.570904 | 0.815818 | 2.076143 | 0.207081 | 0.202172 | 0.050657 | 0.208993 | 0.182306 | 0.00053 |
| (Hex)2 (Fuc)1 + (Man)3(GlcNAc)2 | 0.322479 | 0.213651 | 0.423122 | 0.268242 | 0.046893 | 0.010074 | 0.112891 | 0.049903 | 0.017103 | 0.095448 |
| (Hex)3 + (Man)3(GlcNAc)2 | 4.861187 | 5.775936 | 6.654761 | 5.079474 | 0.305603 | 0.3547 | 1.417819 | 1.179548 | 0.027682 | 0.213128 |
| (Hex)1 (HexNAc)1 (Fuc)1 + (Man)3(GlcNAc)2 | 0.750266 | 0.722315 | 0.514881 | 0.608877 | 0.102088 | 0.016991 | 0.017088 | 0.036114 | 0.664285 | 0.015161 |
| (HexNAc)2 (Fuc)1 + (Man)3(GlcNAc)2 | 1.652909 | 3.456976 | 1.353847 | 3.715004 | 0.132951 | 0.654484 | 0.010019 | 0.352926 | 0.009456 | 0.000317 |
| (Hex)1 (HexNAc)2 + (Man)3(GlcNAc)2 | 0.876123 | 0.772922 | 0.646969 | 0.817259 | 0.129494 | 0.084843 | 0.088617 | 0.055944 | 0.312527 | 0.048099 |
| (Hex)4 + (Man)3(GlcNAc)2 | 2.450873 | 2.895117 | 3.097092 | 2.245556 | 0.253136 | 0.10024 | 0.766259 | 0.409485 | 0.04753 | 0.164815 |
| (Hex)2 (HexNAc)1 (Fuc)1 + (Man)3(GlcNAc)2 | 0.819582 | 0.867903 | 0.64262 | 0.728035 | 0.073132 | 0.069334 | 0.04725 | 0.006566 | 0.452941 | 0.036177 |
| (HexNAc)3 (Fuc)1 + (Man)3(GlcNAc)2 | 2.046827 | 2.744773 | 1.385735 | 2.486179 | 0.256854 | 0.248038 | 0.122548 | 0.205141 | 0.027643 | 0.001339 |
| (Hex)5 + (Man)3(GlcNAc)2 | 3.384827 | 2.579308 | 4.207472 | 2.518858 | 0.330947 | 0.236353 | 1.087182 | 0.392432 | 0.026518 | 0.064636 |
| (HexNAc)4 + (Man)3(GlcNAc)2 | 0.498715 | 0.340168 | 0.283597 | 0.26723 | 0.062833 | 0.027843 | 0.245731 | 0.233142 | 0.016187 | 0.937326 |
| (Hex)6 + (Man)3(GlcNAc)2 | 6.567266 | 2.574261 | 7.830155 | 3.542284 | 0.705065 | 0.227602 | 0.927784 | 0.289577 | 0.000733 | 0.001576 |
| (Hex)2 (HexNAc)1 (Fuc)1 (NeuAc)1 + (Man)3(GlcNAc)2 | 0.635073 | 0.537187 | 0.515992 | 0.507301 | 0.037168 | 0.044506 | 0.061995 | 0.051956 | 0.043073 | 0.861418 |

(m/z protein).

SD; standard deviation, man-; control, man+; cartilage stimulated by α-mannosidase.

mice than in control mice (WT and floxed mice). The mice were followed up until 15 months of age to assess the spontaneous development of OA with aging (*Figure 5F*). There were no apparent changes of OA in the knee joints of either mouse genotype at 3 months of age (*Figure 5G*). At 4 months of age, significant changes in OA were detected in the knee joints of the *Fut8* cKO mice. The cartilage surface integrity was no longer maintained at 9 months. These OA changes progressed more in *Fut8* cKO mice than in their WT littermates at 15 months. These histological findings were quantitatively confirmed by the Osteoarthritis Research Society International (OARSI) scores (difference: –2.000; WT

**Table 2.** List of the top 10 deviations from the y=x line.

The values are expressed as relative values of the amount of substance (pmol) calculated based on internal standards to the total amount of N-glycan.

| Glycan structure | Control | Mannosidase | Dissociation |
| --- | --- | --- | --- |
| (Hex)2 + (Man)3(GlcNAc)2 | 0.13433877 | 0.17518289 | 0.04084413 |
| (HexNAc)2 (Fuc)1 + (Man)3(GlcNAc)2 | 0.01345174 | 0.03953942 | 0.02608767 |
| (Hex)5 + (Man)3(GlcNAc)2 | 0.02873745 | 0.01473236 | 0.01400509 |
| (Hex)6 + (Man)3(GlcNAc)2 | 0.03127904 | 0.01857732 | 0.01270172 |
| (HexNAc)3 (Fuc)1 + (Man)3(GlcNAc)2 | 0.01922672 | 0.03108068 | 0.01185396 |
| (Hex)2 (HexNAc)2 (Fuc)1 | 0.01876579 | 0.00792915 | 0.01083663 |
| (Hex)3 + (Man)3(GlcNAc)2 | 0.03565567 | 0.02670789 | 0.00894778 |
| (HexNAc)1 (Fuc)1 + (Man)3(GlcNAc)2 | 0.01320084 | 0.01968082 | 0.00647997 |
| (HexNAc)4 (Fuc)2 + (Man)3(GlcNAc)2 | 0.05688225 | 0.05042987 | 0.00645238 |

p mol relative (%).

versus *Col2a1-Cre; Fut8^flox/flox*, 95% CI for the difference: –3.723 to –0.2771 at 4 months, difference: –2.700; WT versus *Col2a1-Cre; Fut8^flox/flox*, 95% CI for the difference: –4.423 to –0.9771 at 9 months, and difference: –6.700; WT versus *Col2a1-Cre; Fut8^flox/flox*, 95% CI for the difference: –8.423 to –4.977 at 15 months).

## Altered glycosylation of human OA cartilage based on comprehensive glycan analysis

Total glycome profiling of human OA cartilage is shown in *Figure 6A* and *Figure 6—source data 1*. The amount of high-mannose type N-glycan was significantly decreased, the same as previously reported in OA cartilages (*Figure 6A*; *Urita et al., 2011*). Focusing on fucosylation, most of the complex/hybrid type glycans were modified with fucose. The fucosylation that occurs at the innermost N-acetylglucosamine (GlcNAc) of N-glycans could be recognized with a lectin from the mushroom Pholiota squarrosa (PhoSL) and was able to detect the expression of the corefucose structure. Therefore, we suspected that the stainability of PhoSL is increased in OA cartilage. As expected, the PhoSL staining of OA cartilage was significantly increased relative to that of a healthy control (*Figure 6B*). We performed principal component analysis (PCA) on 1110 glycomic data, discriminating OA from healthy cartilages along the first principal component (PC1) axis (*Figure 6C*). Hierarchical cluster analysis showed that clades containing corefucosylated glycans were clustered (*Figure 6D*), and N-13 ((Hex)1 (HexNAc)1 (Fuc)1 + (Man)3(GlcNAc)2) has the largest squared correlation with its cluster component.

## Discussion

The objective of this work was to provide novel and translational insights into the pathogenesis of OA associated with changes in glycan structure. A graphical abstract summarizing our findings is shown in *Figure 7*. To test the hypothesis that structural changes in high-mannose type N-glycans may induce OA, we tested the effect of mannosidase loading on the structure of high-mannose type N-glycans. Consequently, modifying the structure of high-mannose type N-glycans by mannosidase loading induced OA-like histological alterations. Furthermore, the OA-like changes induced by mannosidase were recovered by enzyme removal. These results suggest that structural alterations of high-mannose type N-glycans lead to OA-like histological changes in articular cartilage but do not necessarily cause the pathogenesis of eventual OA. Subsequently, glycoblotting analysis revealed that core-fucose-containing glycans underwent complementary structural alterations per high-mannose type N-glycans. This suggests that the structural change of core-fucose-containing glycans may have some compensatory mechanism for OA-like alterations caused by the structural change of high-mannose type N-glycans. Furthermore, under the regulation of *Fut8* expression, which regulates the synthesis of core fucose, mannosidase was found to induce irreversible OA. The high-mannose/corefucosylation

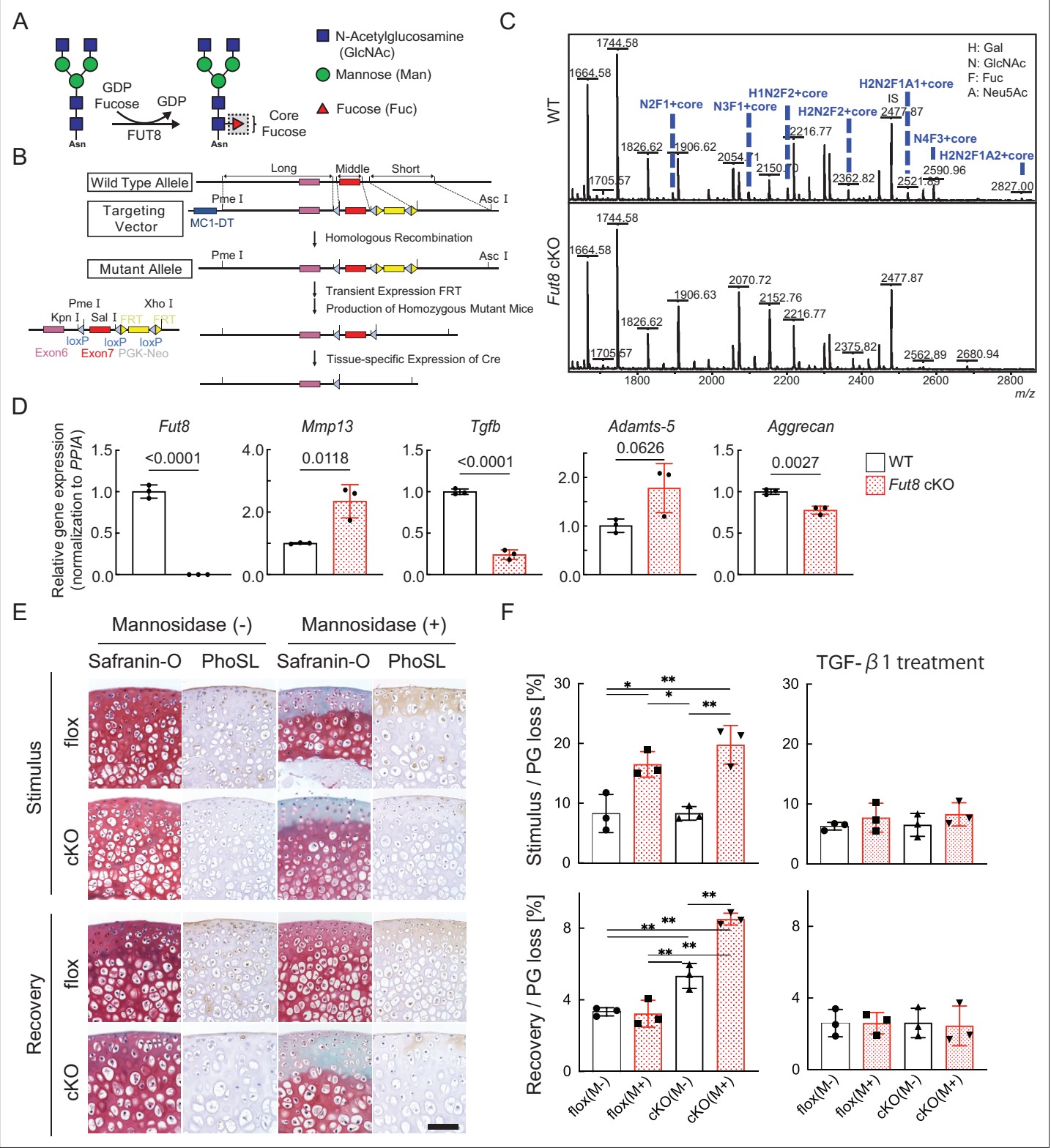

**Figure 4.** Loss of resilience due to FUT8 deficiency in cartilage. (**A**) Biological reaction of FUT8. FUT8 transfers fucose to the innermost GlcNAc residue of complex N-glycans via α1,6-linkage (corefucosylation). (**B**) Targeted disruption of *Fut8* locus. The *Fut8* gene (WT allele; top), targeting vector (middle), and disrupted *Fut8* locus (mutant allele; bottom). Schematic representation of the *Fut8*-targeting strategy and Cre-mediated recombination of the *Fut8*[loxP] allele. (**C**) MALDI-TOF MS mass spectra of N-glycans from WT and *Fut8* cKO mice. The corefucosylation levels in cartilage were decreased and undetectable in *Fut8* cKO mice. (**D**) Gene profile in chondrocytes isolated from *Fut8* cKO mice. The expression levels of these genes in WT cells were

*Figure 4 continued on next page*

*Figure 4 continued*

set to 1. PPIA, peptidylprolyl isomerase A. (**E**) Histological findings in cartilage explants from Fut8 cKO mice and their floxed littermates cultured with mannosidase and subjected to Safranin O staining and PhoSL lectin staining. Scale bar, 50 μm. (**F**) PG release in cultured cartilage explants from *Fut8* cKO mice and their floxed littermates. M-, Mannosidase (-); M+, Mannosidase (+). Data are shown as mean ± standard deviation. In (**F**), n=3 samples (six mice) per group. *p<0.05, **p<0.01. One-way ANOVA, with the Tukey multiple comparison test, was used to perform statistical analysis. MALDI-TOF MS, matrix-assisted laser desorption/ionization-time of flight mass spectrometry; WT, wild-type; cKO, conditional knockout; PG, proteoglycan; ANOVA, analysis of variance.

The online version of this article includes the following figure supplement(s) for figure 4:

**Figure supplement 1.** The proportion of high-mannose type N-glycans in the cartilage of Fut8 cKO mice.

relationship estimated function to maintain formed cartilage. In endochondral ossification, the *Fut8* cKO growth plate had an enlarged hypertrophic zone and reduced primary spongiosa because it is involved in the next process of cartilage replacement into bone rather than the process of cartilage formation. These results provide the first evidence that structural alterations in glycans in the articular cartilage regulate eventual OA development.

Two key targets of cartilage degeneration during OA are type II collagen, a major substrate of matrix metalloproteinase (MMP)–13 (*Wu et al., 2002*), and aggrecan, a PG with glycosaminoglycan side chains of chondroitin and keratin sulfate (*Song et al., 2007*; *Malfait et al., 2002*). The degradation of non-collagenous molecules, such as aggrecan, occurs before the degradation of type II collagen in the early stages of OA (*Malfait et al., 2002*; *Aigner et al., 2001*). The histological analysis in this study revealed that the alteration in the structure of high-mannose type N-glycans resulted in a decrease in Safranin O staining (a decrease in glycosaminoglycans and/or PGs), whereas there was no change in collagen staining. These findings are consistent with alterations in the early stages of OA. Furthermore, we observed that removing mannosidase could cause PG loss to the control level. *Karsdal et al., 2008* demonstrated that articular cartilage degradation is reversible in the presence of aggrecanase and concluded that the repair ability is not impaired before the degradation of aggrecan and type II collagen by MMPs. *Reimann et al., 1982* observed that the glycosaminoglycans loss in articular cartilage is reversible and an early sign of OA before the 'state of no return.' The present results are consistent with the findings of previous reports on early OA and suggest that the mannosidase-loaded model can be considered an early OA-like alteration.

Since the structural change in high-mannose type N-glycans alone did not cause irreversible OA, we investigated other regulatory mechanisms related to high-mannose type N-glycans that could induce irreversible OA. The expression of corefucosylated N-glycans on a series of biosynthetic pathways was upregulated with structural changes in high-mannose type N-glycans. FUT8 is known to corefucosylate small oligomannose N-glycans (Man4-Man5GlcNAc2) but not large oligomannose N-glycans, such as Man8-Man9GlcNAc2 (*Yang and Wang, 2016*). This suggests that small oligomannose N-glycans trimmed by mannosidase could activate *Fut8*. *Fut8* is also upregulated in human OA chondrocytes (*Toegel et al., 2013*; *Yu et al., 2023*), and increased expression of these corefucosylated N-glycans has been observed during late chondrocyte differentiation (*Homan et al., 2019*). These findings suggested that these structural changes are related to OA-related glycosylation.

To clarify the relationship between the alteration of corefucosylated N-glycans and cartilage degeneration, *Fut8* cKO mice were used. Since the results of the ex vivo freeze-thaw model showed that cartilage degeneration was induced by changes in the glycan structure of chondrocytes, we used chondrocyte-specific KO mice. As a result, the reversibility of the cartilage degeneration model was diminished. This finding suggests that *Fut8*-mediated corefucosylation in chondrocytes plays a function in promoting glycan repair. Furthermore, to directly determine whether *Fut8* was involved in the inhibition of OA progression, the OA model was investigated. For the modeling of OA in *Fut8* cKO mice, the instability-induced OA model and the age-associated OA model were adapted. The former emphasizes mechanical stress factors in OA, the latter aging factors. OA is a multifactorial disease. Therefore, we thought it was appropriate to validate both aspects of OA. The results showed that conditional KO of *Fut8* in chondrocytes hastened the onset of OA. This suggests that cartilage corefucosylation associated with FUT8 plays a protective role against the attenuation of type II collagen in the OA process.

Finally, the expression of α1,6-linked core fucose on N-glycans in human cartilage is associated with OA, and corefucosylation was found to be one of the characteristics in OA cartilage. Total cellular

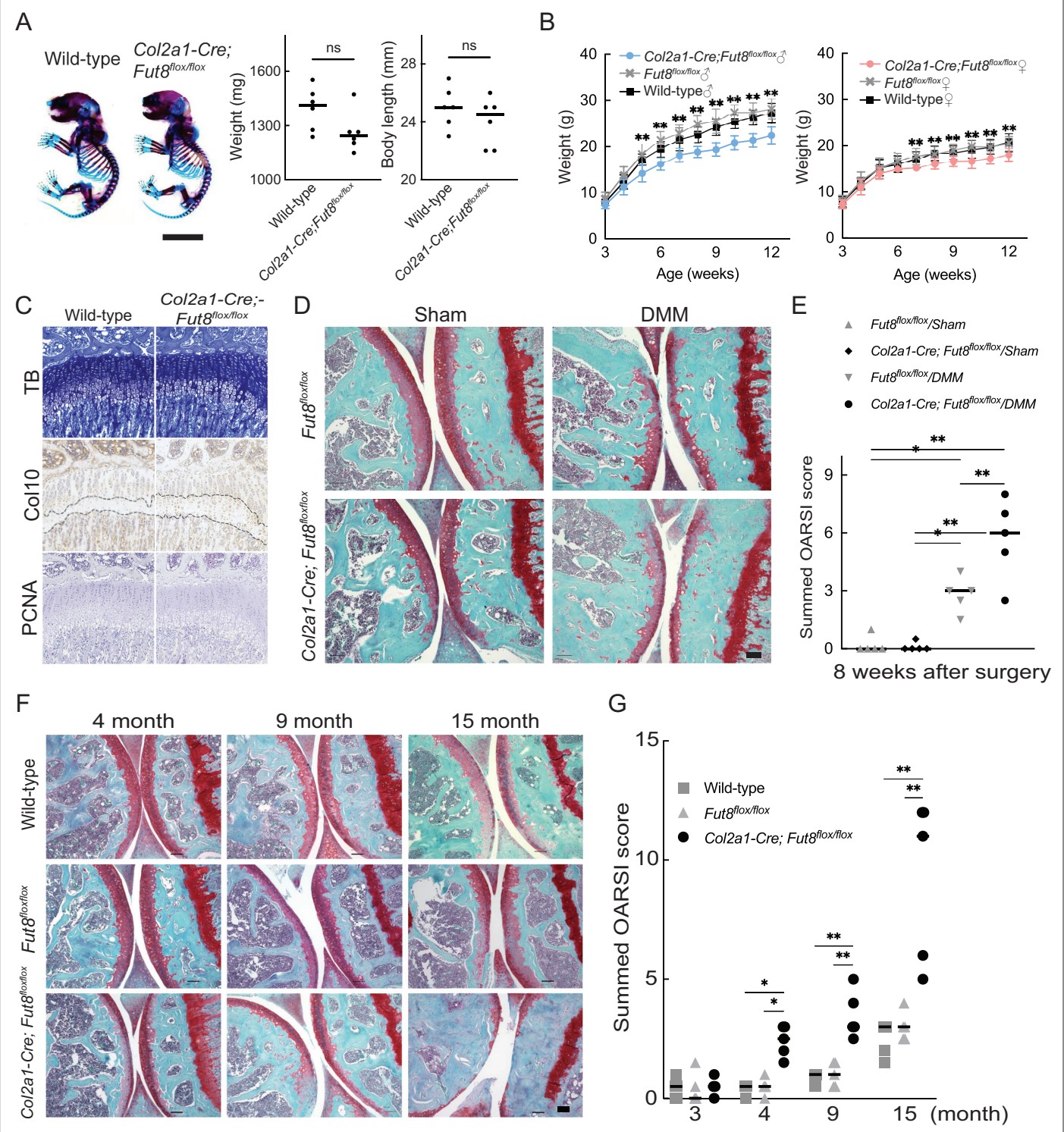

**Figure 5.** OA acceleration in *Fut8* cKO mice. (**A**) Double staining with alizarin red and alcian blue of the whole skeleton of wild-type and *Col2a1-Cre;Fut8^flox/flox* cKO littermate embryos (newborn). Scale bars, 1 cm (left). Weight and body length of wild-type and cKO littermate embryos (right). (**B**) Growth curves were determined by body weight in male (left) and female (right) wild-type mice and their cKO littermates. (**C**) Histologic findings in knee joints from 4-week-old mice. Toluidine blue (TB) staining and immunostaining for type X collagen (Col10) and proliferating cell nuclear antigen (PCNA) in growth plates of the tibia were performed in each mouse genotype. The dashed line indicates the hypertrophic zone. Scale bar, 100 μm. (**D**), (**E**) Features of instability-induced OA in *Fut8^flox/flox* (flox) mice and their cKO littermates 8 weeks after surgery. Safranin O staining is shown for each mouse genotype. Scale bar, 100 μm (**D**). Summed histological scores for OA severity in the knee cartilage from flox and cKO mice, as determined using

*Figure 5 continued on next page*

*Figure 5 continued*

the OARSI scoring system, are shown (**E**). (**F**), (**G**) Features of age-associated osteoarthritis in wild-type mice and their flox and cKO littermates. Safranin O staining of the knee joint is shown for each mouse genotype at 3, 4, 9, and 15 months of age. Scale bar, 100 µm (**F**). The summed OARSI scores are shown (**G**). Data are shown as mean ± standard deviation. In (**A**), the Welch t-test was used to perform statistical analyses (n=6 mice per group). In (**B**), n=15 mice per group at each time point. In (**E**) and (**G**), n=5 mice per group at each time point. *p<0.05, **p<0.01 versus the wild-type group in (**B**) and (**G**), and versus flox mice in (**E**). One-way ANOVA with the Tukey multiple comparisons test (**B**), and two-way ANOVA with the Tukey multiple comparisons test (**E and G**) were used to perform statistical analysis. OA, osteoarthritis; cKO, conditional knockout; OARSI, Osteoarthritis Research Society International. ANOVA, analysis of variance; ns, not significant.

The online version of this article includes the following figure supplement(s) for figure 5:

**Figure supplement 1.** Expanded zone of hypertrophic chondrocytes in Fut8 cKO mice.

glycome analysis, including N- and O-linked glycans derived from glycoproteins, GSL-glycans, GAGs, and fOSs, is informative for analyzing cell-specific characteristics (*Fujitani et al., 2013*; *Yoshida et al., 2016*; *Furukawa, 2017*). Glycome profiling enabled the detection of corefucosylated N-glycans by sequential clustering of degenerated cartilage characteristics. Core fucose deficient mice generated by ablating the α1,6 fucosyltransferase enzyme, FUT8, were reported to have suppressed phosphorylation of Smad and increased expression of MMPs (matrix metalloproteinases) due to decreased binding of TGF-β ligand caused by the lack of core fucose addition to the TGF-β type II receptor (*Gao et al., 2012*; *Wang et al., 2005*). The present study shows that treatment of cartilage with mannosidase caused corefucosylation, suppression of *Mmp13*, and increased expression of *Tgfb*. Moreover, treatment of FUT8-knockout cartilage with mannosidase resulted in the suppression of *Tgfb* expression and increased *Mmp13* expression. Here, the exogenous addition of TGF-β1 rescued them from cartilage degeneration. These results suggest that corefucosylation following the loss of high-mannose type N-glycans may provide chondroprotective effects via the TGF-β signaling pathway, which may have been lost by FUT8-knockout.

One limitation of this study was that we did not have access to cartilage samples from immediately before the onset of OA; thus, we do not know the expression of Fut8 in human cartilage in the early stage of OA onset and its influence on extracellular matrix degradation under homeostatic conditions. Obtaining cartilage specimens continues to be challenging, especially for those with early stages of OA.

In summary, we show for the first time a key role of FUT8 and glycan-dependent signaling mediators in extracellular matrix resilience associated with cartilage corefucosylation during early OA.

## Methods

### Intra-articular injection of α-mannosidase in the rabbit knee

All animal experiments were performed in accordance with the Japanese Act on Welfare and Management of Animals. Our animal use protocol was approved by the Institutional Animal Care and Use Committee of Hokkaido University (Approval No. 17–0059). We injected mannosidase into rabbit knee joints following a previous paper showing that N-type glycans are altered prior to cartilage matrix degeneration. Japanese White rabbits weighing 3.10±0.15 kg purchased from a professional breeder (Japan SLC Inc, Hamamatsu, Japan) were used for this study according to established ethical guidelines approved by the local animal care committee. Animals were anesthetized with 10 mg/kg of intravenous ketamine, followed by isoflurane in oxygen gas. Both knees in each rabbit were shaved, prepared, and draped in a sterile fashion. α-Mannosidase from Canavalia ensiformis (Jack bean; Sigma, St Louis, MO, USA) was dissolved in saline. Five hundred microliter volumes containing 1.9 units per ml (0.05 mg) of mannosidase solution were injected into the joint cavity of the right knee. α-Mannosidase from Canavalia ensiformis (Jack bean) has been widely used as a tool for glycan analysis, which has a high specificity towards terminal mannosidic residues of glycans and participates in mannose trimming reactions (*Gonzalez and Jordan, 2000*). The measurement of its performance in a saline solution was evaluated using the QuantiChrom α-Mannosidase Assay Kit (BioAssay Systems, Hayward, CA, USA). This colorimetric assay uses 4-nitrophenyl-α-d-mannopyranoside as a substrate to identify the generation of 4-nitrophenol. The following doses of α-Mannosidase were administered: 0.5 ml at 0.01 µg/ml, 1.0 µg/ml, or 100 µg/ml. Simultaneously, the contralateral knee joint received an equal volume of saline solution. Each joint was injected once or four times in alternate weeks. The

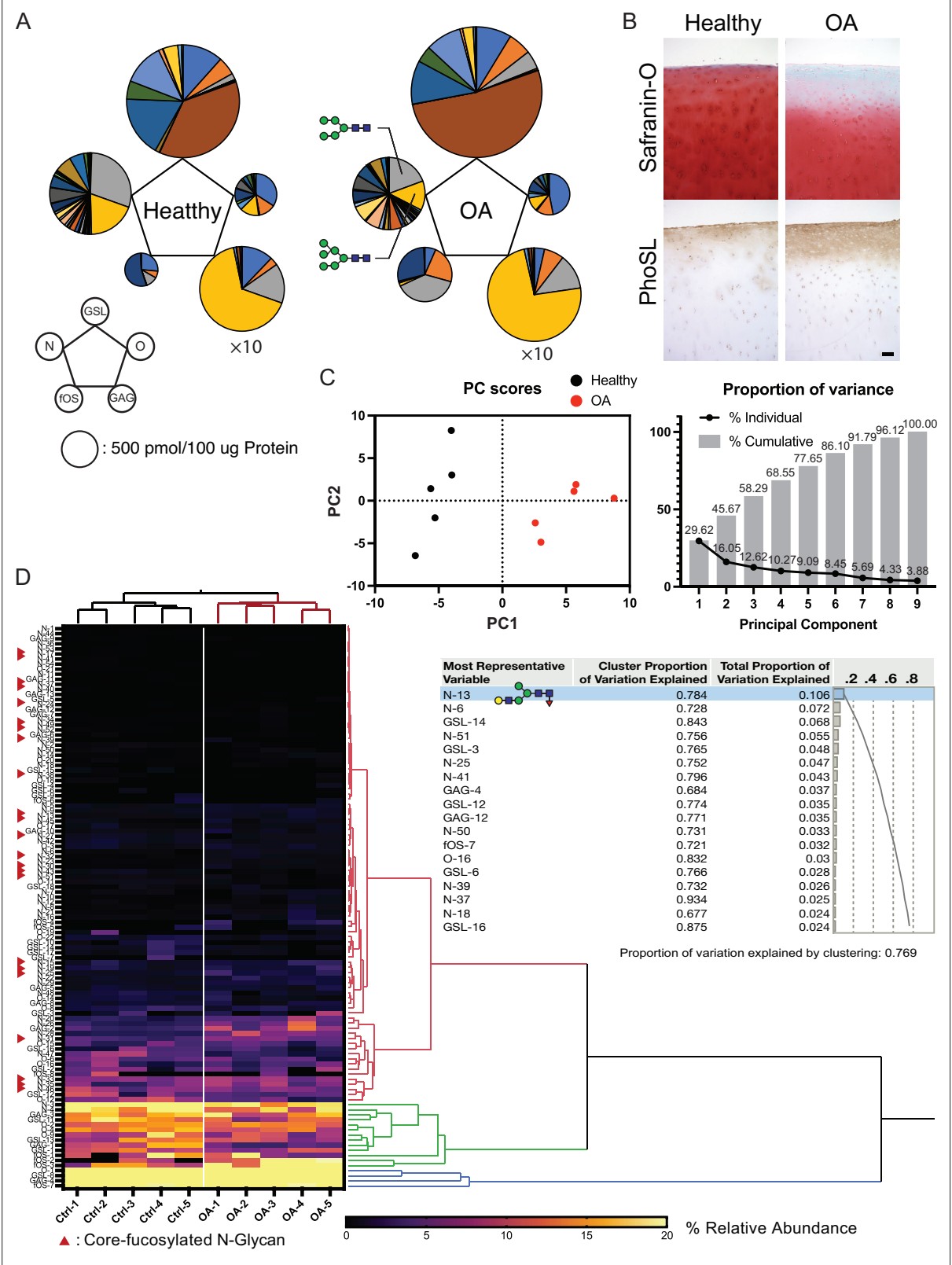

**Figure 6.** Altered glycosylation of human Osteoarthritis (OA) cartilage based on comprehensive glycan analysis. (**A**) Total glycome profiling of human OA cartilage. Pie charts at the vertices of the pentagon correspond to the glycan expression profiles of N-glycans, O-glycans, GSL-glycans, free oligosaccharides (fOS), and glycosaminoglycan (GAG). The size of each circle and its constituent colors reflect the absolute quantity of glycans (pmol/100 μg protein) and the glycan substructures, respectively. The sizes of the circles representing the GAG contents are decreased by 1/10-fold.

*Figure 6 continued on next page*

*Figure 6 continued*

Each color indicates the estimated glycan structure and corresponds to the respective glycan number in the ***Figure 6—source data 1***. (**B**) Expression of core fucose in healthy and OA cartilage. Scale bar, 100 μm. (**C**) Principal component analysis (PCA) of a glycan expression data set. Data points represent individual samples. The first principal component (PC1) distinguishes healthy and OA samples. (**D**) Hierarchical cluster analysis results showing cluster image display for total glycans with color gradient for relative glycan expression and dendrogram for each glycan structure. Cluster summary of PCA on the glycome are shown on the right. The most representative variable is (Hex)1 (HexNAc)1 (Fuc)1 + (Man)3(GlcNAc)2 which means the largest squared correlation with its cluster component.

The online version of this article includes the following source data for figure 6:

**Source data 1.** List of glycans quantified in healthy and osteoarthritic human cartilage.

animals were euthanized at 4 and 16 weeks postoperatively. Each section of the knee joint was stained with hematoxylin and eosin or Safranin O/fast green, as well as with immunohistochemical staining for type II collagen and TUNEL. Two investigators quantified the OA severity using the OARSI scoring method.

## Evaluation of collagen orientation by a polarized light microscope

To evaluate the collagen orientation of the repaired tissue at 16 weeks postoperatively, a supplemental evaluation using a polarized light microscope (PLM; ECLIPSE E600 POL; Nikon, Tokyo, Japan) was conducted (*Ross et al., 2013*). This evaluation was performed on HE-stained sections from each group. Two cross polarizers were used so that highly ordered collagen perpendicular to the articular surface appeared bright, while collagen that was not ordered (nonbirefringent) appeared dark. The contrast of fibrils more parallel to the articular surface was darker than collagen oriented perpendicular to the articular surface. To observe the predominant direction of the birefringent regions and confirm the lack of orientation in the nonbirefringent regions, the sections were rotated at 0°, 45°, and 90° for the fixed filters. Microscopic images were acquired using a digital camera (DS-5M-L1, Nikon, Tokyo, Japan). Each sample was examined independently by an experienced PLM user.

## Ex vivo analysis of deletion of high-mannose type N-glycan in cartilage

Organ culture experiments in mice were established to study the effects of mannosidase on articular cartilage without immunoreaction and in anticipation of later candidate gene research using transgenic mice. According to the established ethical guidelines approved by the local animal care committee, cartilage catabolism was analyzed by culturing femoral head cartilage of 4-week-old C57BL6 male mice with α-mannosidase (1.9 U/mL) ex vivo (*Glasson et al., 2005*; *Stanton et al., 2011*). Mannosidase was removed on day 3, and organ culture was continued until day 6. To evaluate cartilage degradation, quantification of PG release from cartilage explants was performed using the

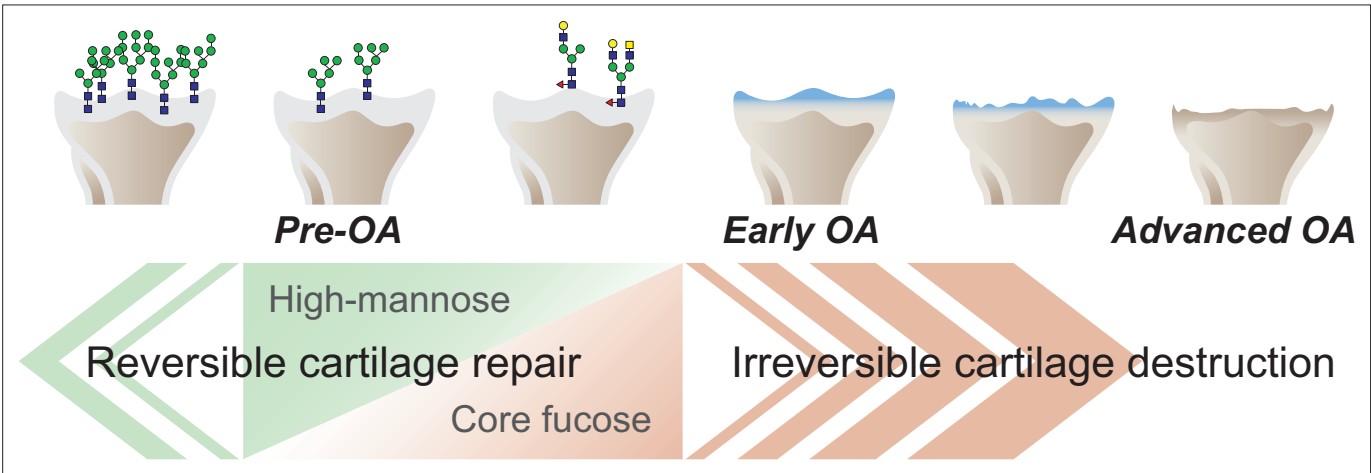

**Figure 7.** Graphical abstract. The corefucosylation of N-glycans is associated with the reduction of high-mannose type N-glycans in articular cartilage functions to maintain the formed vitreous cartilage as cartilage. Failure of the complementary relationship leads to irreversible cartilage degeneration and destruction of cartilage integrity.

dimethylmethylene blue assay (*Stanton et al., 2011*; *Estes et al., 2010*) and exposure to sialidase (1.9 U/mL) and interleukin-1 beta (Cat #. 300–01 A; BioLegend, San Diego, CA, USA; 10 ng/mL) was also used as another control. The NO concentration in the culture medium was measured using the Griess reagent system (Promega, Tokyo, Japan). To determine the direct effects of mannosidase on the matrix, some cartilage explants were subjected to freeze-thaw cycles in liquid nitrogen to destroy chondrocytes present in the superficial layer of the cartilage (*Karsdal et al., 2008*; *Patwari et al., 2004*). After the procedure, cartilage explants were replaced with the same culture medium described above. Each tissue sample was stained with Safranin O. Lectin staining with Con A (Wako Pure Chemical Industries, Richmond, VA, USA) and PhoSL (J-Oil Mills Inc, Tokyo, Japan) was used to detect the high-mannose type N-glycans. Staining was performed using the ABC Kit (Vector Labs, Burlingame, CA, USA), and the sections were counterstained with HE.

## RNA extraction and quantitative RT-PCR analysis

The chondrocytes isolated from 6-day-old mice were seeded onto a 24-well plate and stimulated by treatment with α-mannosidase (1.9 U/mL) for 74 hr. Total RNA was extracted from the samples using an RNeasy Mini Kit (QIAGEN, Hilden, Germany). For complementary DNA synthesis, 1.0 µg of RNA was reverse transcribed using random hexamer primers (Promega, Tokyo, Japan) and ImProm II reverse transcriptase (Promega). Quantitative RT-PCR was performed using a Thermal Cycler Dice Real Time System II (Takara Bio Inc, Otsu, Japan) and SYBR Premix Ex Taq II (Takara, Shiga, Japan) with gene-specific primers (*Supplementary file 1*) in 15 µl of the mixture, following the manufacturer's instructions. Quantitative data were normalized using peptidylprolyl isomerase A as an endogenous reference gene and were calculated using the ΔΔCt method (*Rao et al., 2013*).

## Generation of *Fut8^{loxp/loxp}Col2a1-Cre* cKO mice

*Fut8^{loxp/loxp}* mice were provided by Tohoku Pharmaceutical University (Sendai, Japan). To interrupt the synthesis of corefucosylated N-glycans in cartilage, we generated mice with knockout of the chondrocyte-specific *Fut8* gene (*Col2a1;Fut8^{-/-}* mice) by crossing *Fut8^{loxp/lox}* mice with Col2a1-Cre–transgenic mice (*Ovchinnikov et al., 2000*), in which Cre recombinase is expressed specifically in chondrocytes under the Col2a1 promoter. Transgenic mice carrying the Col2a1-Cre transgene (strain B6; SJL-Tg [Col2a1-Cre] 1Bhr/J; stock number 003554) were obtained from the Jackson Laboratory (Bar Harbor, ME, USA). Homozygous *Fut8^{loxp/loxp}* mice were bred with *Col2a1-Cre* mice, and offspring with the genotype *Fut8^{loxp/+} Col2a1-Cre* were bred with *Fut8^{loxp/loxp}* mice to obtain the *Fut8^{loxp/loxp}Col2a1-Cre* cKO experimental group and *Fut8^{loxp/loxp}* (flox) littermate controls. All animals were maintained at Hokkaido University under the regulations of the Institutional Animal Care and Use Committee.

## Skeletal examination

To observe the skeletal systems, whole skeletons of WT and cKO littermates were stained with alcian blue and alizarin red, as previously described (*Yamada et al., 2006*; *Nakamichi et al., 2003*). The newborns were skinned, eviscerated, and fixed in 95% ethanol. Intact skeletons were stained with 150 mg/ml of alcian blue 8 GX in 75% (vol/vol) ethanol/20% acetic acid for 24–48 hr and excess tissues were removed by 1–2% KOH digestion for 2–3 days. The skeletal preparations were stained with 75 mg/ml of alizarin red S in 1% KOH for 24 hr and cleared in graded glycerol solutions (*Lufkin et al., 1992*).

## Instability-induced OA model

An OA model was created in 10-week-old mice by destabilizing the knee joint as previously described (*Matsui et al., 2009*). With the mice under general anesthesia, we destabilized the right knee joint by transection of the medial collateral ligament and removing the cranial half of the medial meniscus using a microsurgical technique (DMM). A sham operation was performed on the left knee joint using the same approach, without ligament transection and meniscectomy. For histological assessment, the mice were sacrificed, and the entire knee joint was dissected 8 weeks postoperatively.

### Age-associated OA model

At 3, 4, 9, and 15 months of age, the mice were sacrificed, and their entire knee joints were dissected to assess the spontaneous development of OA as a model of age-associated OA (*Matsui et al., 2009*; *Stoop et al., 1999*).

### Human cartilage preparation

Acquisition and use of patient tissues were approved by the institutional review board (IRB) of Hokkaido University (approval number: 014–0144), and informed consent was obtained in advance. Samples were from patients undergoing total knee arthroplasty for clinically and radiologically diagnosed OA (n=5; age range 61–84 years, mean 72.4; male: female ratio 1:4). Age-matched control articular cartilage was from patients undergoing total hip arthroplasty for femoral neck fracture with no history of joint disease (n=5; age range 45–88 years, mean 61.2 years; male: female ratio 3:2). Each cartilage specimen was evaluated macroscopically by at least two veteran doctors to ensure that it had not suffered damage or degeneration and was treated within 9 hr of harvest in the operating room. Immediately upon receipt, isolated human cartilage was homogenized with a Polytron blender (Kinematica, Luzern, Switzerland), and then cold ethanol was added to separate the protein and lipid fractions.

### Total cartilage glycomic analysis

To determine whether the glycosylation detected is conserved across species, we analyzed the total glycome in human cartilage. N-glycans were released directly using cartilage lysates following deglycosylation by overnight treatment with peptide N-glycanase F (PNGase F, 2 U) (Roche, Switzerland). The supernatants containing GSLs and fOSs were dried with a centrifugal evaporator. GSL-glycans were isolated by enzymatic digestion using EGCase I (Takara Bio Inc Japan), whereas fOSs were recovered from an EGCase I-free fraction (*Fujitani et al., 2011*). Cartilage lysate was delipidated and digested by GAG disaccharides (*Takegawa et al., 2011*). Extracted glycoproteins were subjected to BEP for O-glycan analysis. Detailed procedures and materials are provided elsewhere (*Furukawa et al., 2015*). N-glycans, fOSs, GAGs, and GSL-glycans were subjected to glycoblotting (*Furukawa et al., 2008*). Purified N-glycans, fOSs, O-glycans, and GSL-glycan solutions were mixed with 2,5-dihydrobenzoic acid and subjected to Matrix-assisted laser desorption/ionization-time of flight mass spectrometry (MALDI-TOF MS). 2AB-labeled GAG disaccharides were analyzed by HPLC. The glycan compositions were manually determined by conducting database searches (i.e. a compositional search of the UniCarbKB database (http://www.unicarbkb.org/query) for fOSs and N- and O-glycans and of the SphinGOMAP database (http://www.sphingomap.org/) for GSL-glycans). All previously deposited GSL-glycans in the SphinGOMAP database were extracted and compiled as an in-house database to allow searching by the m/z value and/or composition. The absolute quantification was obtained by comparative analyses between the MS signal areas derived from each glycan and the internal standard.

### Statistics

All data in this study are presented as the mean ± standard deviation and were repeated at least three times unless otherwise indicated. Data analysis was performed using GraphPad Prism 9 software version 9.4.1 (GraphPad Software, Inc, San Diego, CA, USA), and the Welch *t*-test/Welch analysis of variance with subsequent use of the Tukey–Kramer multiple comparison tests were used to determine significant differences between the groups. p-values < 0.05 were considered significant. JMP Pro version 16.2.0 (SAS Institute, Cary, NC, USA) was used to reveal the relationships between each subject of sub glycomes; we performed principal component analysis (PCA) and hierarchical cluster analysis.

## Acknowledgements

We thank the Iwasaki laboratory members for technical support, J Gu (Tohoku Pharmaceutical University) for the Flox-α1,6 fucosyltransferase (*Fut8*) mouse, Y Kobayashi (J-OIL MILLS, Inc) for PhoSL-lectin. This work was funded by the Japan Society for the Promotion of Science KAKENHI (Grant-in-Aid for Challenging Exploratory Research: funding numbers: 16K15650, 19K22672 [to NI]; Grant-in-Aid

for Research (B): funding numbers: 22H03917 [to TO]; and Grant-in-Aid for Early-Career Scientists: funding number: 19K18516 [to KH]) and AMED (Grant Number JP22zf01270004h0002 [to NI, TO, KH]).

## Additional information

### Funding

| Funder | Grant reference number | Author |
|---|---|---|
| Japan Society for the Promotion of Science | 16K15650 | Norimasa Iwasaki |
| Japan Society for the Promotion of Science | 19K22672 | Norimasa Iwasaki |
| Japan Society for the Promotion of Science | 22H03917 | Tomohiro Onodera |
| Japan Society for the Promotion of Science | 19K18516 | Kentaro Homan |
| Japan Agency for Medical Research and Development | JP22zf01270004h0002 | Kentaro Homan Norimasa Iwasaki Tomohiro Onodera |

The funders had no role in study design, data collection and interpretation, or the decision to submit the work for publication.

### Author contributions

Kentaro Homan, Funding acquisition, Investigation, Visualization, Methodology, Writing - original draft, Project administration, Writing - review and editing; Tomohiro Onodera, Supervision, Funding acquisition, Investigation, Methodology, Writing - original draft, Project administration, Writing - review and editing; Hisatoshi Hanamatsu, Investigation, Visualization, Writing - review and editing; Jun-ichi Furukawa, Validation, Investigation, Writing - review and editing; Daisuke Momma, Masatake Matsuoka, Investigation, Writing - review and editing; Norimasa Iwasaki, Conceptualization, Supervision, Funding acquisition, Investigation, Project administration, Writing - review and editing

### Author ORCIDs

Kentaro Homan (ORCID) http://orcid.org/0000-0002-0288-7691
Tomohiro Onodera (ORCID) http://orcid.org/0000-0002-4308-174X

### Ethics

Acquisition and use of patient tissues were approved by the institutional review board (IRB) of Hokkaido University (approval number: 014-0144), and informed consent was obtained in advance.
All animal experiments were performed in accordance with the Japanese Act on Welfare and Management of Animals. Our animal use protocol was approved by the Institutional Animal Care and Use Committee of Hokkaido University (Approval No. 17-0059). All surgery was performed under ketamine anesthesia, and every effort was made to minimize suffering.

Reviewer #1 (Public Review): https://doi.org/10.7554/eLife.92275.3.sa1
Reviewer #2 (Public Review): https://doi.org/10.7554/eLife.92275.3.sa2
Author Response https://doi.org/10.7554/eLife.92275.3.sa3

## Additional files

### Supplementary files

- Supplementary file 1. A primer list of qRT-PCR.
- MDAR checklist

- Source data 1. Raw data for all figures.

## Data availability

All data generated or analyzed during this study are included in the manuscript and source data files.

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
