## [Editor Report · eLife assessment]

This **valuable** study sheds light on the pivotal role of alterations in chondrocyte glycan metabolism in two contexts: The onset of cartilage degeneration and early onset of osteoarthritis (OA). The action is through hypertrophic differentiation of chondrocytes, a finding that provides insights into the identification of nascent markers for early-stage OA. The evidence supporting the claims is **solid**, with the authors clearly demonstrating the role of articular cartilage corefucosylation in the development of OA. The authors' inferences would be further enhanced through future experiments aimed at analyzing the mechanisms underlying the changes in glycometabolism in cartilage.

---

## [Referee Report · Reviewer #1 (Public Review)]

Summary:

This study is valuable in that it may lead to the discovery of future OA markers, etc., in that changes in glycan metabolism in chondrocytes are involved in the initiation of cartilage degeneration and early OA via hypertrophic differentiation of chondrocytes. However, more robust results would be obtained by analyzing the mechanisms and pathways by which changes in glycosylation lead to cartilage degeneration.

Strengths:

This study is important because it indicates that glycan metabolism may be associated with pre-OA and may lead to the elucidation of the cause and diagnosis of pre-OA.

Weaknesses:

More robust results would be obtained by analyzing the mechanism by which cartilage degeneration induced by changes in glycometabolism occurs.

---

## [Referee Report · Reviewer #2 (Public Review)]

Summary:

This paper consists of mostly descriptive data, judged from alpha-mannosidase-treated samples, in which they found an increase in core fucose, a product of Fut 8.

Strengths:

This paper is interesting in the clinical field, but unfortunately the data is mostly descriptive and does not have a significant impact on the scientific community in general.

Weaknesses:

If core fucose is increased, at least the target glycan molecules of core fucose should be evaluated. They also found an increase in NO, suggesting that inflammatory processes also play an important role in OA in addition to glycan changes.

It has already been reported that core fucose is decreased by administration of alpha-mannosidase inhibitors. Therefore, it is expected that alphaa-mannosidase administration increases core fucose.

---

## [Author Response]

The following is the authors’ response to the original reviews.

**eLife assessment**
This study is valuable as it sheds light on the pivotal role played by alterations in glycan metabolism within chondrocytes in the onset of cartilage degeneration and early onset of osteoarthritis (OA) through the process of hypertrophic differentiation of chondrocytes, giving insights into the identification of nascent markers for early-stage OA. Although the methods, data, and analyses broadly support the claims, the data shown by the authors are incomplete because the mechanism by which cartilage degeneration induced by changes in glycometabolism occurs has not been fully elucidated. The authors' deductions stand to gain further credence through undertaking additional experiments aimed at analyzing the mechanisms underlying the changes in glycometabolism in cartilage, such as the meticulous identification of the target glycan molecules bearing core fucose and analysis of endochondral ossification in cartilage-specific Fut8 KO mice.

We wish to express our strong appreciation to the Reviewer for his or her insightful comments on our paper. We feel the comments have helped us significantly improve the paper. In particular, we wish to acknowledge the Reviewer’s highly valuable comments on the effect of Fut8 on endochondral ossification.

**Reviewer #1 (Public Review):**:Summary:This study is valuable in that it may lead to the discovery of future OA markers, etc., in that changes in glycan metabolism in chondrocytes are involved in the initiation of cartilage degeneration and early OA via hypertrophic differentiation of chondrocytes. However, more robust results would be obtained by analyzing the mechanisms and pathways by which changes in glycosylation lead to cartilage degeneration.Strengths:This study is important because it indicates that glycan metabolism may be associated with pre-OA and may lead to the elucidation of the cause and diagnosis of pre-OA.

We thank reviewer #1 for their interest in our work and their overall positive report.

Weaknesses:More robust results would be obtained by analyzing the mechanism by which cartilage degeneration induced by changes in glycometabolism occurs.

To understand the mechanisms of cartilage degeneration induced by changes in glycometabolism, we attempted additional experiments using rescue experiments with external administration of TGF-β. We had shown that the addition of mannosidase to an organ culture system of normal wild-type mouse cartilage increased TGF-β gene expression from 6 hours (Fig. 3E) and that TGF-β expression was even suppressed in chondrocytes from Fut8 cKO mice (Fig. 4D). In addition to these results, an early OA model in which mannosidase is added to the cartilage was used to test the effect of exogenous TGF-β. As a result, under TGF-β treated conditions, no degenerative changes occurred when high-mannose type N-glycans were trimmed, and proteoglycan leakage during the recovery period was significantly reduced. This was considered to be a very useful finding and it was decided to include the experimental results in Figure 4F, rather than making them supplement data.

**Reviewer #2 (Public Review):**
Summary:This paper consists of mostly descriptive data, judged from alpha-mannosidase-treated samples, in which they found an increase in core fucose, a product of Fut 8.Strengths:This paper is interesting in the clinical field, but unfortunately, the data is mostly descriptive and does not have a significant impact on the scientific community in general.

We thank reviewer #2 for their interest in our work and their overall positive report. In response to your comment about our attempts to show that glycan changes occur at the precursor stage of cartilage substrate degeneration and that this glycosylation is also what triggers substrate degeneration, we would like to add that reversing cartilage substrate degeneration is a very ambitious challenge. We are currently in the preparatory stages of characterizing the appropriate glycan-substrate relationships to 'rescue' cartilage tissue from degeneration, and we hope to use this approach to provide information on the pre-developmental stages of OA.

Weaknesses:If core fucose is increased, at least the target glycan molecules of core fucose should be evaluated. They also found an increase in NO, suggesting that inflammatory processes also play an important role in OA in addition to glycan changes.

As the increase in NO was observed in the organ culture system and cartilage is a tissue without vascular invasion, we thought that the involvement of immune cells could be excluded. On the other hand, our research group has reported that chondrocytes themselves have inflammatory circuits (Ota et al., Arthritis Rheum. 2019. DOI:10.1002/art.41182), but as we did not find increased expression of NF-κB, an indicator of inflammatory amplifier activation, we concluded that inflammation was not involved in this study.

It has already been reported that core fucose is decreased by administration of alpha-mannosidase inhibitors. Therefore, it is expected that alpha-mannosidase administration increases core fucose.

The report by Toegel et al. that the synthesis of complex-type N-glycans (Man2a1, Mgat2) is predicted in human OA chondrocytes along with the expression of Fut8 also led to the expectation that administration of α-mannosidase would increase core fucose. However, there was no conclusive evidence that administration of α-mannosidase increased core fucose; in 1987, Vignon et al performed an enzyme assay on experimental OA cartilage (rabbit ACLT model) and showed that mannosidase was very high in operated joints and that its activity increased and decreased with the severity of fibrosis in the cartilage. The results suggest that glycoprotein hexose degradation is an early transient event in the enzymatic process of cartilage destruction. These findings led to the conception of a novel 'pre-OA model' in which mannosidase is added to the joint. The present study is valuable in its demonstration that glycometabolism is a driver of degeneration.

(see manuscript REF. 25, 9)

Toegel et al., Arthritis Res. Ther. 2013. DOI:10.1186/ar4330

Vignon et al., Clin Rheumatol. 1987. DOI:10.1007/BF02201026

**Reviewer #3 (Public Review):**
Summary:In the manuscript "Articular cartilage corefucosylation regulates tissue resilience in osteoarthritis", the authors investigate the glycan structural changes in the context of pre-OA conditions. By mainly conducting animal experiments and glycomic analysis, this study clarified the molecular mechanism of N-glycan core fucosylation and Fut8 expression in the extracellular matrix resilience and unrecoverable cartilage degeneration. Lastly, a comprehensive glycan analysis of human OA cartilage verified the hypothesis.Strengths:Generally, this manuscript is well structured with rigorous logic and clear language. This study is valuable and important in the early diagnosis of OA patients in the clinic, which is a great challenge nowadays.

We thank reviewer #3 for their interest in our work and their mainly positive report. This is precisely the purpose of our study, as we are primarily interested in the detection of conditions prior to the onset of OA.

Weaknesses:I recommend minor revisions:(1) I would suggest the authors prepare an illustrative scheme for the whole study, to explain the complex mechanism and also to summarize the results.

We would like to thank the reviewer for this comment and have created a new Figure 7 for the overall study scheme.

We included the following statement in the opening discussion part:

"The objective of this work was to provide novel and translational insights into pathogenesis of OA associated with changes in glycan structure. A graphical abstract summarizing our findings is shown in Fig. 7." (line199-201, p9)

(2) Including but not limited to Figures 2A-C, Figures 3A and C, Figure 4B, and Figures 5A and D. The texts in the above images are too small to read, I would suggest the authors remake these images.

The font size of the figures has been reviewed and revised throughout.

(3) The paper is generally readable, but the language could be polished a bit. Several writing errors should be realized during the careful check.

Thanks to your suggestion, I have noticed several writing errors. In addition, we have had the manuscript rewritten by an experienced scientific editor, who has improved the grammar and stylistic expression of the paper.

(4) As several species and OA models were conducted in this study, it would be better if the authors could note the reason behind their choice for it.

The authors agree with the reviewer's argument that since several species and OA models were performed in this study, it would be better to note the reason for their choice.

We first attempted to inject mannosidase into rabbits, matching the animal species to a previous paper showing that N-glycans are altered prior to degeneration of the cartilage matrix. Next, we checked whether similar changes occur in mouse cartilage after mannosidase treatment, assuming that we would verify this in genetically engineered mice. We then used the integrated glycome in human cartilage to see if the corefucosylation phenomenon detected was conserved across species.

For the modeling of OA in Fut8 cKO mice, the instability-induced OA model and the age-associated OA model were adapted. The former emphasizes mechanical stress factors in OA, the latter aging factors. OA is a multifactorial disease. Therefore, we thought it was appropriate to validate both aspects of OA.

We included the following statements in each Methods part:

"We injected mannosidase into rabbit knee joints in accordance with a previous paper showing that N-type glycans are altered prior to cartilage matrix degeneration." (line289-290, p12)

"Organ culture experiments in mice were established to study the effects of mannosidase on articular cartilage without immunoreaction and in anticipation of later candidate gene research using transgenic mice." (line326-328, p14)

"To determine whether the glycosylation detected is conserved across species, we analyzed the total glycome in human cartilage." (line407-408, p17)

We included the following statements in the Discussion part:

"For the modeling of OA in Fut8 cKO mice, the instability-induced OA model and the age-associated OA model were adapted. The former emphasizes mechanical stress factors in OA, the latter aging factors. OA is a multifactorial disease. Therefore, we thought it was appropriate to validate both aspects of OA." (line254-257, p11)

**Reviewer #1 (Recommendations For The Authors):**
(1) The cited literature states that core fucosylation by FUT8 has a chondroprotective effect via the TGF-β pathway and that the loss of these chondroprotective effects in Fut8 led to cartilage degeneration, but these need to be proven by experiment.

We agree that corefucosylation and the TGF-β signaling pathway are important lines of investigation. We have now acknowledged this and added in the revised manuscript that additional experiments have shown that TGF-β restores the protective effects of Fut8 cKO cartilage by external administration.

We included the following statements in the Results part:

"To evaluate whether TGF-β1 decreases cartilage degeneration after mannosidase stimulation, TGF-β1 was exogenously added to Col2-Fut8−/− cartilage in the presence of α-mannosidase stimulation for 24 h. The samples treated with TGF-β1 leaked significantly less PG following mannosidase stimulation compared to samples not treated with TGF-β1 (Fig. 4F)." (line143-147, p6-7)

We included the following statements in the Discussion part:

"Here, the exogenous addition of TGF-β1 rescued them from cartilage degeneration." (line274-275, p12)

(2) There are skeletal differences in cartilage-specific Fut8 KO mice compared to WT, and the effect of Fut8 on endochondral ossification should also be analyzed.

We agree that Fut8 is associated with various endochondral ossification processes (for example by the TGF-β signaling pathway). Moreover, we would like to thank the reviewer for the proposed experiment.

The growth curve was normal at birth, with differences beginning around weaning (~3 w for mice). Therefore, we evaluated the epiphyseal line of 4-week-old mice stained with toluidine, type 10 collagen, and proliferating cell nuclear antigen. This is similar to the epiphyseal growth plate phenotype of Smad3ex8/ex8 mice by Yang et al. and is consistent with the finding that Smad3 deficiency does not affect chondrogenesis during developmental stages, but the hypertrophic zone is increased in 3-4 week-old Smad3 KO mice. Chondrocytes in Fut8 cKO mice were suppressed of Tgf-β expression (Fig. 4D), suggesting that inhibition of TGF-β signaling, which is suppressive for late hypertrophic chondrocyte differentiation, led to the increased height of the hypertrophic zone.

The results suggested that the growth plate of Fut8 cKO mice had an enlarged hypertrophic layer and decreased primary trabecular bone. Because these results have important implications for the content of the paper, we have included the staining results in Figure 5 and added a graph quantitatively assessing the extent of the hypertrophic zone as supplementary Figure S6.

We included the following statement in the Results part:

"To assess the role of FUT8 in endochondral ossification, we performed an epiphyseal plate analysis of 4-week-old Col2-Fut8−/− mice. This uncovered a significant enlargement of the zone of hypertrophic chondrocytes in the growth plates of the long bones of Col2-Fut8−/− mice compared to controls (Fig. 5C, S6 Figure)." (line154-158, p7)

We included the following statement in the Discussion part:

"The high-mannose/corefucosylation relationship estimated function to maintain formed cartilage. In endochondral ossification, the Fut8 cKO growth plate had an enlarged hypertrophic zone and reduced primary spongiosa because it is involved in the next process of cartilage replacement into bone rather than the process of cartilage formation." (line214-217, p9)

Literature mentioned above (not included in manuscript):

Yang X, et al. TGF-beta/Smad3 signals repress chondrocyte hypertrophic differentiation and are required for maintaining articular cartilage. J Cell Biol. 2001;153(1):35–46.

(3) The DMM model analysis is performed with n=5 for each group. Please consider if the sample size is sufficient.

In the literature, the sample sizes for DMM models have varied in previous studies (Doyran et al., n=5; Liao et al., n=6-7; Ouhaddi et al., n=8). Therefore, we performed a preliminary test of the DMM in WT and Flox mice with n=3 each and a power analysis with the outcome set to the OARSI score at 8 weeks. This resulted in n=4. The sample size for this study was increased to n=5 to account for attrition. The summed OARSI score of the WT in this study was comparable to that of Ouhaddi et al. and the model was judged to be working accurately. The summed OARSI score of the WT in this study was comparable to that of Ouhaddi et al. and the model was judged to be working accurately. The summed OARSI score of the WT in this study was comparable to that of Ouhaddi et al. and the model was judged to be working accurately.

Literature mentioned above (not included in manuscript):

(1) Doyran B, Tong W, Li Q, Jia H, Zhang X, Chen C, et al. Nanoindentation modulus of murine cartilage: a sensitive indicator of the initiation and progression of post-traumatic osteoarthritis. Osteoarthr Cartil. 2017;25(1):108–17.

(2) Liao L, Zhang S, Gu J, Takarada T, Yoneda Y, Huang J, et al. Deletion of Runx2 in Articular Chondrocytes Decelerates the Progression of DMM-Induced Osteoarthritis in Adult Mice. Sci Rep. 2017 24;7(1):2371.

(3) Ouhaddi Y, Nebbaki SS, Habouri L, Afif H, Lussier B, Kapoor M, et al. Exacerbation of Aging-Associated and Instability-Induced Murine Osteoarthritis With Deletion of D Prostanoid Receptor 1, a Prostaglandin D2 Receptor. Arthritis Rheum. 2017;69(9):1784–95.

**Reviewer #2 (Recommendations For The Authors):**
This paper is suitable for publication in clinical Journals related to osteoarthritis and cartilage.Identification of core fucosylated glycans from chondrocytes is essential for this type of paper.

We mentioned that we had identified similar corefucosylated glycans in isolated mouse chondrocytes from the cartilage (line117-118, p5), but we have now also added the following to the subtitle of the Results section to avoid any potential confusion: "Corefucosylated N-glycan was formed in resilient cartilage and its isolated chondrocyte" (line109, p5)

Thank you again for your comments on our paper. We trust that the revised manuscript is suitable for publication.